# Closed-Loop Control of Melt Pool Temperature during Laser Metal Deposition

**DOI:** 10.3390/s24155020

**Published:** 2024-08-02

**Authors:** Qing Wang, Jinchao Zhang, Qingqing Zhu, Yue Cao

**Affiliations:** 1School of Information Science and Technology, Nantong University, Nantong 226019, China; wq990117@163.com (Q.W.); 18252509379@163.com (Q.Z.); 2School of Optoelectronic Science and Engineering, Soochow University, Suzhou 215021, China; 3Nantong Dart-Rich Fan Co., Ltd., Nantong 226019, China

**Keywords:** laser metal deposition, closed-loop control, temperature control, PI controller

## Abstract

Laser metal deposition (LMD) is a technology for the production of near-net-shape components. It is necessary to control the manufacturing process to obtain good geometrical accuracy and metallurgical properties. In the present study, a closed-loop control method of melt pool temperature for the deposition of small Ti6Al4V blocks in open environment was proposed. Based on the developed melt pool temperature sensor and deposition height sensor, a closed-loop control system and proportional-integral (PI) controller were developed and tested. The results show that with a PI temperature controller, the melt pool temperature tends to the desired value and remains stable. Compared to the deposition block without the controller, a flatter surface and no oxidation phenomenon are obtained with the controller.

## 1. Introduction

Laser metal deposition (LMD) is an additive manufacturing (AM) technology of metal materials where a laser beam is used to melt powders onto a substrate. It can directly manufacture or repair near-net-shape parts with complex geometries [1,2,3]. Moreover, it allows for the fabrication of functionally graded materials and multifunctional homogenous or heterogeneous structures [4,5]. It has been shown to significantly reduce lead times and production costs.

Titanium alloys have been widely used in aerospace, biomedicine, and marine engineering fields, due to their high specific strength, excellent biocompatibility, and fine corrosion resistance [6,7,8]. LMD technology provides a new method for the manufacturing of titanium alloy parts with high performance and complex structure. However, titanium alloys easily react with atmospheric gases at high temperatures during the LMD process, which reduces the morphology and properties of deposited parts [9,10]. Therefore, most LMD processes of titanium alloys are produced in an inert gas-sealed chamber [11]. However, this method is not suitable for manufacturing large-size parts or in situ repair of components. In addition, due to the low thermal conductivity of titanium alloy, heat accumulation occurs during the deposition process, which affects the forming accuracy and the forming quality of parts [12,13,14].

For the control of oxidation out of chamber, a local inert gas environment generated by a shielding nozzle, which focuses only on the deposition region, is an alternative to a sealed chamber. It can not only reduce gas consumption but also increase the nozzle flexibility to fabricate large parts and in situ repair. Ding et al. [15] designed a local shielding nozzle to prevent atmospheric contamination. Computational fluid dynamics models were used to verify the possibility of wire + arc additive manufacture (WAAM) of reactive materials out of chamber. Bermingham et al. [16] investigated the sensitivity of Ti6Al4V parts to oxidation contamination during the WAAM process out of chamber when using an inert gas-shielding nozzle with different configurations. In previous studies [17,18], a coaxial double-layer shielding gas nozzle was developed based on annular laser metal deposition (ALMD). The results showed that ALMD-produced Ti6Al4V out of chamber was viable. The deposited components presented a silver-white luster, indicating negligible oxidation. However, with the increase in deposition time, especially for small parts, the high-temperature region increased due to heat accumulation. The local inert gas area generated by the shielding nozzle cannot completely protect the melt pool, resulting in oxidation during the process [19]. Thus, it is necessary to control the parameters in the deposition process.

LMD involves complex interactions among laser, powder, and substrate in a local inert gas environment. Fluctuations in processing parameters such as melt pool temperature and layer height could result in defects in the deposited samples [20]. It is desirable to have a real-time monitoring and control system to guarantee deposition quality. The melt pool temperature is an important processing parameter, which is related to the geometrical characteristics and microstructure properties. Bernauer et al. [21] used a dedicated closed-loop control system based on pyrometer signals to control the melt pool temperature. It was found that the melt pool temperature was positively correlated with the width and dilution of the single track. Petrat et al. [22] investigated the effect of the travel path strategy and position on temperature evolution. The results showed that the zigzag strategy had the smallest maximum temperatures and the smallest temperature differences in the edge area. Devesse et al. [23] introduced a control system based on a heat conduction model. A PI controller was designed to regulate the melt pool width by adjusting the laser power. The controller can be used for different AM processes by modifying the signal-processing strategies. Mazzarisi et al. [24] analyzed the entire thermal field produced by the LMD process in different deposition strategies using an IR thermal camera. An ad hoc algorithm was developed to determine thermal characteristics to analyze metallographic and microhardness. 

The deposition height is another important processing parameter, which affects the forming precision. Donadello et al. [25] presented a height-monitoring system based on optical triangulation. The measurement error signal was employed in feedback control systems to control the process parameters to adapt to the actual deposition growth. A stainless steel cylinder with high precision was successfully manufactured. Yu et al. [26] used machine vision to monitor deposition height in real-time, and proposed a novel self-attention temporal convolutional network (SA-TCN) to predict future deposition height. The prediction accuracy reached an outstanding 99.71%, which can accurately control the height of parts. The above research mainly focus on single-parameter control, which is suitable for handling geometrically complex parts. A multivariable control is desired to be implemented, through which the melt pool temperature, as well as deposition height, are adjusted simultaneously. Song et al. [27] adopted three high-speed cameras and a dual-color pyrometer to monitor the melt pool height and melt pool temperature, respectively. The hybrid control of the height controller and temperature controller obtained stable layer growth and improved geometrical accuracy. However, an LMD of reactive alloys in an open environment based on a closed-loop control system has not been reported.

In this work, a closed-loop melt pool temperature control system for the deposition of the Ti6Al4V alloy in an open environment is developed. A double-color infrared pyrometer is used to monitor the melt pool temperature. A CCD-based sensor is used to monitor the deposition height and to guide the lift amount of the robot. A PI temperature controller is designed to control melt pool temperature by adjusting laser power at each layer. This controller is able to reduce heat accumulation through heat input control. In addition, the lift amount of the robot corresponds to the deposition height. A small Ti6Al4V block is manufactured using the closed-loop control system, and the deposition quality is analyzed.

## 2. Experimental Setups

### 2.1. Annular Laser Metal Deposition (ALMD)

The ALMD system was used for this study, as shown in Figure 1. It included a 2 kW continuous-wave fiber laser (IPG YLS-2000-TR), a six-axis robot (KUKA KR 60-3F), a powder feeder (GTV PF2/2), and a self-developed annular-beam powder-feeding nozzle [28]. The annular laser beam was obtained by beam shaping [29,30]. The powder stream sprayed by a single powder-feeding tube was surrounded by the annular laser. A developed coaxial double-layer shielding device was used to generate a local protective atmosphere to prevent oxidation.

Gas-atomized spherical Ti6Al4V powder with particle size 75–106 µm was used as feedstock powder. The pure titanium rolled plate was used as the substrate. Before ALMD experiments, the powder was dried in a vacuum oven for 2 h at 120 °C, and the substrate surface was cleaned with acetone.

### 2.2. Closed-Loop Melt Pool Temperature Control System

A closed-loop melt pool temperature control system was developed, as shown in Figure 2. A double-color infrared pyrometer and a high-speed CCD camera were connected to the ALMD nozzle, which monitored the melt pool temperature and the deposition height, respectively. In order to better monitor the melt pool, the monitoring angle of the pyrometer and the CCD camera to the laser beam were 30° and 45°, respectively.

The wavelengths of the pyrometer were 1.4–1.6 µm and 1.6–1.8 µm, and the measured temperature was calculated by Kirchhoff law. The pyrometer had a measuring range from 800 °C to 3000 °C with an accuracy of ±10 °C. As the middle point of the melt pool was used as the monitoring point, the temperature of the molten pool changed little during the deposition process, which was caused by temperature distributions of the annular beam that increased the temperature at the edges and weakened at its center [31]. In order to obtain a large temperature fluctuation, a position 2.5 mm behind the melt pool was selected as temperature monitoring point. The values of melt pool temperature were transmitted in real-time to the industrial computer with a baud rate of 115,200 value/s through serial communication. Then the values of laser power (*P*) were processed by the temperature control algorithm and transmitted to a robot control cabinet by TCP communication. Finally, the robot control cabinet controlled the fiber laser (IPG YLS-2000-TR) through I/O signals.

The deposition height was obtained by image processing with a frame rate of 33 fps, and calculated in the industrial computer. The deposition height of each layer was processed by the height control algorithm and transmitted to a robot control cabinet [32]. The lift amount (ΔZ) was set equal to the deposition height of each layer, which was controlled by a KUKA robot.

## 3. Digital PI Controller Design and Test

In order to ensure the stability of melt pool temperature, the temperature control between layers is designed. $\bar{T}$ is the average melt pool temperature of a layer, where n is temperature points in one layer, $T(t_{j})$ is the actual measured temperature at the time point $t_{j}$ (j=1, 2, …, n). If the process parameters vary with $T(t_{j})$ at each time point, the deposition layer may present defects due to non-uniform heating in a layer. Thus, the process parameters are adjusted only once after a layer to maintain a constant melt pool temperature.
(1)$$\bar{T}=\frac{\sum_{j=1}^{n}T(t_{j})}{n}$$

The e(k) represents the temperature error between the desired temperature $T_{r}(k)$ and the actual temperature $\bar{T}(k)$ of melt pool in the *k*th layer. The *k* is the current layer number.
(2)$$e\left(k\right)=T_{r}\left(k\right)-\bar{T}\left(k\right)$$

In empirical models of the ALMD process, a higher laser power *P* can provide more laser energy to the melt pool and produce a higher melt pool temperature. Inversely, a lower laser power *P* reduces the melt pool temperature. Therefore, the temperature error e(k) can be diminished by adjusting the laser power *P* in each layer.

The PID controller has been widely applied in the industry process control system, due to its simple structure, convenient adjustment, and good stability. When the parameters of the controlled plant are not quite certain or an accurate mathematical model cannot be obtained, the PID controller can still achieve stable control. Therefore, a PI control algorithm was designed:(3)$$\Delta P\left(k\right)=K_{P}e\left(k\right)+K_{I}\sum_{m=1}^{k}e\left(m\right)$$
where ∆P is the increment of the laser power between two adjacent layers, $K_{P}$ is the proportional gain, and $K_{I}$ is the integral gain.
(4)$$P\left(k+1\right)=P\left(k\right)+\Delta P(k)$$
where P(k) is the laser power of the current layer and P(k+1) is the laser power of the next layer. Combining Equations (3) and (4), the PI controller can be written as
(5)$$P\left(k+1\right)=P\left(k\right)+K_{P}e\left(k\right)+K_{I}\sum_{m=1}^{k}e\left(m\right)$$
where the proportional term $K_{P}e(k)$ makes the melt pool temperature of the next layer close to the desired melt pool temperature. The integral term $K_{I}\sum_{m=1}^{k}e\left(m\right)$ can eliminate the accumulated temperature error during the ALMD process.

Figure 3 shows the block diagram of the closed-loop temperature control with laser power as the single control variable. In order to maintain the defocus distance constant in each layer, that is, the laser spot size remains constant, the lift amount (Δ*Z*) is in accord with the measured deposition height of each layer, which is carried out together with the temperature controller in Equation (5).

The step response test of the PI controller was performed, as shown in Figure 4. A thin-wall part with a length of 20 mm and a height of 40 layers was manufactured. The one-way scanning strategy was employed. The scanning speed was 6 mm/s, the defocus distance was −4 mm, the powder feeding rate was 1.4 g/min, the internal and external shielding gas flow rates were 18 L/min and 12 L/min, respectively, and the initial laser power was 900 W. In order to ensure the integrity of the deposited layers, the lower limit of laser power was set to 500 W, and the upper limit of that was 1200 W. The control parameters $K_{P}$ and $K_{I}$ were 1.05 s^−1^ and 0.0242 s^−1^, respectively.

In Figure 4, the desired melt pool temperature changed two times: 1450 °C→1660 °C. The melt pool temperature and the laser power vary with the number of layers, as shown in Figure 5. In the 1–20th layer, the desired melt pool temperature was set as 1450 °C and the PI controller in Equation (5) was activated at the second layer. After the 7th layer, the melt pool temperature was controlled to be stable at the desired temperature. The steady-state fluctuation range of the temperature was ±15 °C, and the laser power varied from 500 W to 690 W. In the 21–40th layer, the desired melt pool temperature rose to 1660 °C. The melt pool temperature stabilized after the 27th layer, and the steady-state fluctuation range of the temperature was ±20 °C.

## 4. Small Block Deposition of Ti6Al4V Alloy

### 4.1. Comparison between Deposited Layers with and without Closed-Loop Control

In the previous study [18], the Ti6Al4V alloy was prepared in an open environment, using the ALMD process with a coaxial double-layer shielding device. However, for the deposition of small structure parts, when the fixed process parameters are employed, the heat accumulation is high due to the continuous thermal input and slow heat dissipation, which easily leads to the oxidation of titanium alloys while processing in an open environment, and reduces the forming quality of the deposited parts [33]. In order to reduce the heat accumulation, a PI closed-loop temperature control system can be used to control melt pool temperature, and to prevent oxidation.

Two small blocks with four passes and fifteen layers were manufactured with and without temperature closed-loop control. The length was 20 mm, and the overlapping ratio of the multi-pass layer was 40%. For the temperature closed-loop control experiment, the initial process parameters and control parameters were the same as those in Section 3. The desired melt pool temperature was set as 1120 °C. The strategy of the melt pool temperature control for the small block was as follows: a layer consisting of four passes was used as a period of regulation. Each single pass in one layer was deposited using the same laser power and overlapping ratio. After the current layer was deposited, the average temperature of the middle two passes in the current layer was calculated, which was regarded as the temperature of the current layer. Then the laser power of the next layer was adjusted based on the control model and reference temperature. In addition, according to the measured deposition height of the current layer, the deposition nozzle was lifted at the same value. Another comparative test without temperature control was designed, that is, the block deposition was carried out in constant laser power during the ALMD process.

Figure 6 shows the melt pool temperature during the deposition process without a controller. The curve of melt pool temperature of the 9th layer is shown in Figure 7. As can be seen from Figure 7, the melt pool temperature of the first and fourth passes fluctuates greatly (grey curves), which is mainly related to the paraxial temperature monitoring method. The elliptic temperature monitoring area cannot all be located on the upper surface of the side pass, which will result in measurement errors. Therefore, the average temperature of the middle two passes (blue curves) is chosen as the temperature of the current layer. As can be seen from Figure 6, with the increase in deposition time, the melt pool temperature increases from 927 °C to 1201 °C, due to the effect of heat accumulation. The melt pool temperature for the first five layers rises rapidly, and goes up slowly for the rest layers. The increase in melt pool temperature will expand the high temperature region. When the high-temperature region exceeds the gas protective area generated by the double-layer shielding device, titanium alloys will be oxidized.

Figure 8 shows the temperature-change curve during the entire deposition process with a PI temperature controller. The desired melt pool temperature of each layer was set as 1120 °C. With the increase in the deposition layers, the melt pool temperature gradually converges to the desired value. The PI temperature controller can control the melt pool temperature well, make it tend to the desired value, and then keep it stable. The constant temperature of the melt pool can keep the high-temperature zone unchanged to ensure that the high-temperature zone is effectively protected in a local inert atmosphere during the whole deposition process.

Figure 9 shows the melt pool temperature of each layer with a PI temperature controller and the corresponding laser power. As can be seen from Figure 9, the melt pool temperature in the first layer is relatively low, due to a faster heat dissipation in contact with the substrate. At the second layer, the PI controller is started and the laser power gradually rises. With the increase in laser power, the melt pool temperature gradually rises and tends to the desired value. When the 5th layer is reached, the laser power increases to 1000 W and the melt pool temperature reaches 1170 °C, exceeding the desired value. After feedback, the laser power begins to decrease, so that the melt pool temperature falls back to near the desired temperature of 1120 °C. At the 8th layer, the melt pool temperature reaches the steady-state value. The PI temperature controller adjusted the laser power dynamically to stabilize the melt pool temperature to 1120 °C. The steady-state fluctuation range of the temperature is ±30 °C. It shows that the PI temperature controller can stabilize the melt pool temperature near the desired value by controlling the laser power dynamically.

### 4.2. Analysis of the Deposition Quality

The manufactured small Ti6Al4V blocks with and without closed-loop control are shown in Figure 10. In Figure 10a, with constant laser power, the surface of the deposited block turns from silver to gold, and to dark blue from the bottom to the top. According to an American Welding Society standard (AWS D17.1) [34], titanium or titanium alloy will be discolored by atmospheric pollution and heat, and the color gradually becomes darker with increasing oxidation. The silver and yellow are regarded as the acceptable colors of the oxidized titanium alloy [35]. The heat accumulation becomes worse with the increasing of deposition layers due to poor heat dissipation for small volumes. As can be seen from Figure 6, with the increase in deposited layers, the melt pool temperature gradually rises and the high-temperature area continues to expand. When the high-temperature region exceeds the effective protection range generated by the shielding device, the oxidation of titanium alloy will occur during deposition. Thus, as the number of deposited layers increases, oxidation becomes more severe and the surface color becomes darker.

As can be seen from Figure 10b, with closed-loop control, the whole surface of the deposited block presents bright silver. It indicates that the deposited layers are adequately shielded during the whole ALMD process, and no oxidation occurs. From Figure 8, the melt pool temperature is stabilized around the desired value. Laser power is dynamically adjusted to prevent excessive heat accumulation during the deposition process, and thus the high-temperature region remains stable and can be adequately shielded by an inert atmosphere.

In addition, it can be seen that without closed-loop control, the top surface of the block is uneven due to the instability of the melt pool caused by excessive heat accumulation. In the incomplete shielding condition, oxides accumulate layer by layer. Oxides can change the wetting and spreading of the melt pool, which will impact the melt pool dimension and the wetting angle [9]. In addition, oxides have limited adhesion to the matrix and may break due to the accumulation of thermal stress. These may cause defects and prevent further deposition on the part, and lead to uneven layer surface. For a small block with closed-loop control, the top surface is flatter and smoother than the one fabricated without closed-loop control. This is because complete shielding prevents the high-temperature region from oxidation, thereby reducing the amount of oxides and providing a stable condition for successive deposition [9].

The cross-section was extracted for macrostructure observation by an optical microscope (OM). Metallographic samples were prepared by standard mechanical polishing method and etched with Kroll reagent. The OM macrostructures of deposited small blocks with and without closed-loop control are shown in Figure 11. The β columnar grains are observed in both samples, which are growing epitaxially across layers and driven by heat flux toward the substrate. The outline of the melt pool can be clearly noticed, and no obvious metallurgical defects are observed. The “dark” and “bright” β grains appear because of different crystal orientations [36,37]. It is noted that the average width of the β columnar grains prepared with closed-loop control is smaller than that without closed-loop control, which is mainly due to the difference in heat input.

Microhardness of two samples was measured from top to bottom by a MH-5 Vickers tester using a load of 0.5 kg and loading time of 15 s, as shown in Figure 12. It can be seen that the microhardness of the blocks is between 300 HV and 425 HV. However, in the top region, the microhardness of the block without closed-loop temperature control is higher than that with closed-loop control. It is mainly related to the increase in oxygen content in the block without closed-loop control, which is consistent with the above-mentioned result that the top surface presents blue where oxidation occurs. The chemical composition of the upper surface on the blocks with and without control was analyzed by energy dispersive spectrometry (EDS), and the results are shown in Table 1. It is found that the oxygen level of the uncontrolled block is higher than that of controlled block. Oxygen atoms dissolved in titanium lattice can cause lattice distortion, which can enhance the deformation resistance and thus increase the microhardness.

## 5. Conclusions

In this paper, a closed-loop melt pool temperature control system dedicated the ALMD process was presented. The following conclusions can be drawn:
In the ALMD technology, a closed-loop control method of melt pool temperature for the deposition of small blocks in an open environment was proposed. Compared with the traditional constant parameter deposition method, a layer with variable parameters can reduce heat accumulation, and improve deposition quality.A double-color infrared pyrometer was developed to measure the melt pool temperature. A PI temperature controller was designed to control melt pool temperature by adjusting the laser power after each layer. A CCD-based sensor was used to monitor the deposition height, and the lift amount remained the same as the deposition height.The step response test indicated that the PI temperature controller had a stable and convergent control performance. With a PI controller, a small block with no oxidation was successfully manufactured in an open environment. The melt pool temperature tended to the desired value and the heat accumulation was reduced. Compared to the deposition results without the controller, a flatter surface and no oxidation phenomenon were obtained with the controller.


## Figures and Tables

**Figure 1 sensors-24-05020-f001:**
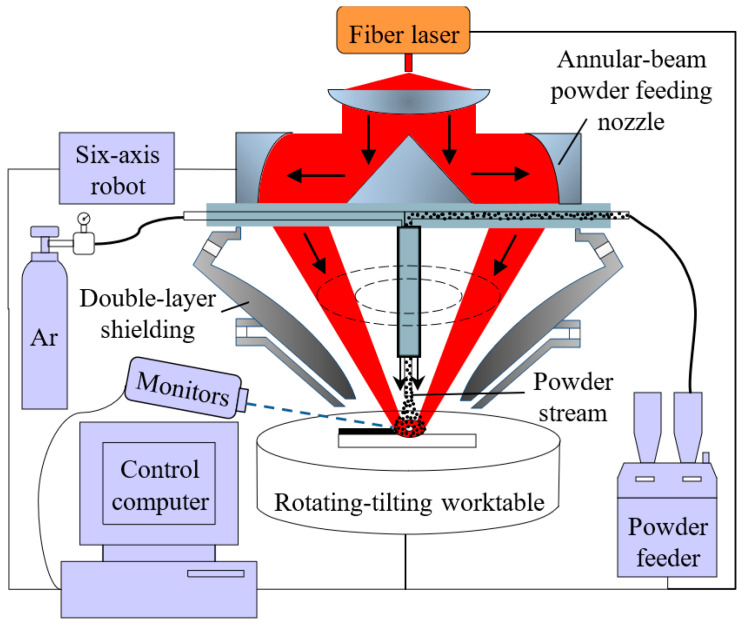
The schematic diagram of ALMD system.

**Figure 2 sensors-24-05020-f002:**
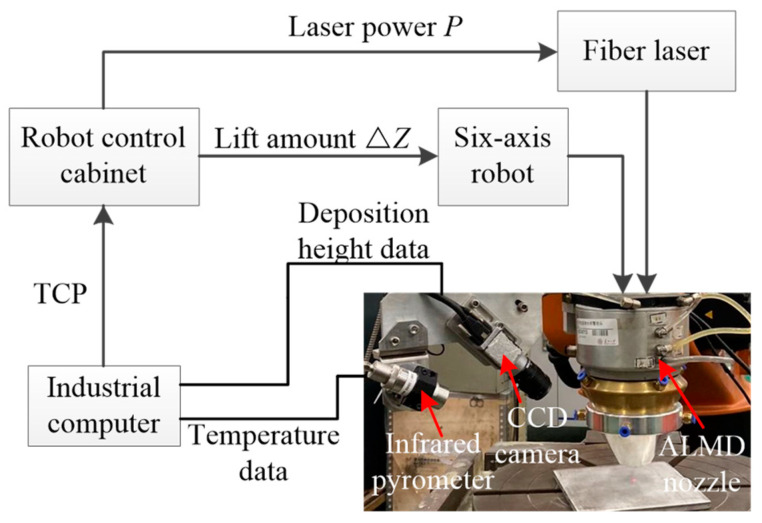
Methodology for closed-loop control of ALMD process.

**Figure 3 sensors-24-05020-f003:**
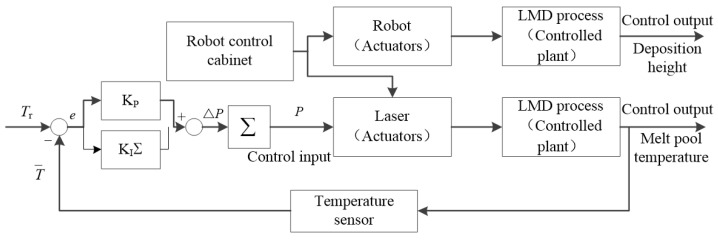
The block diagram of the temperature control.

**Figure 4 sensors-24-05020-f004:**
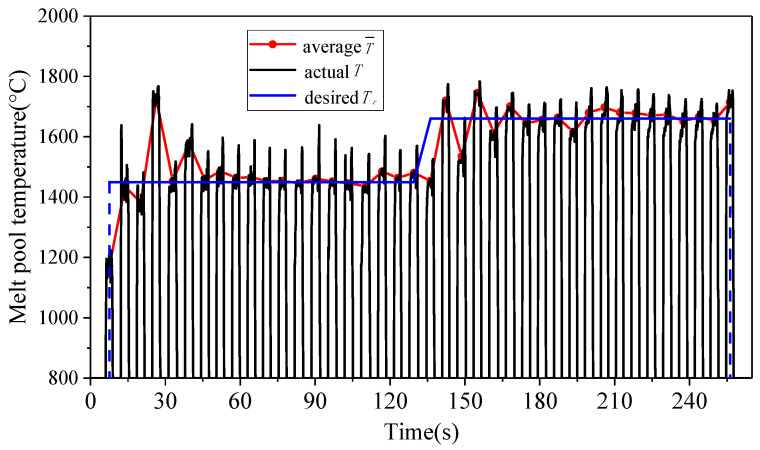
Step responses under the step signals of the desired melt pool temperature.

**Figure 5 sensors-24-05020-f005:**
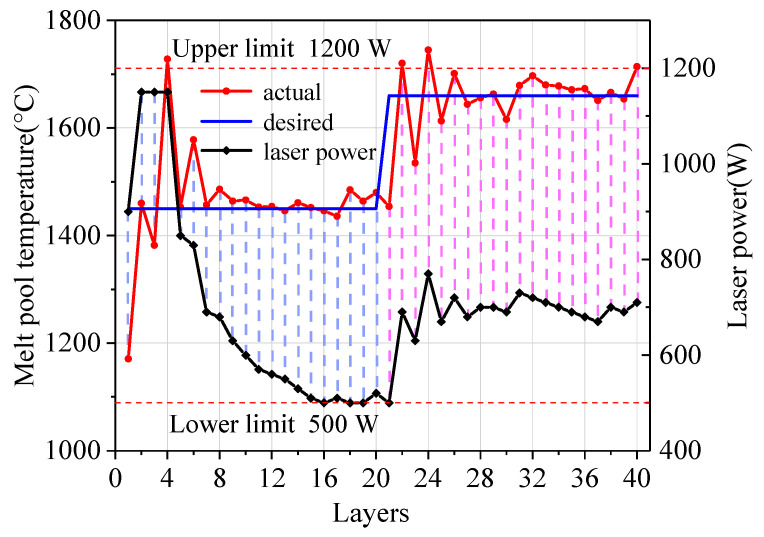
The melt pool temperature and laser power under different layers.

**Figure 6 sensors-24-05020-f006:**
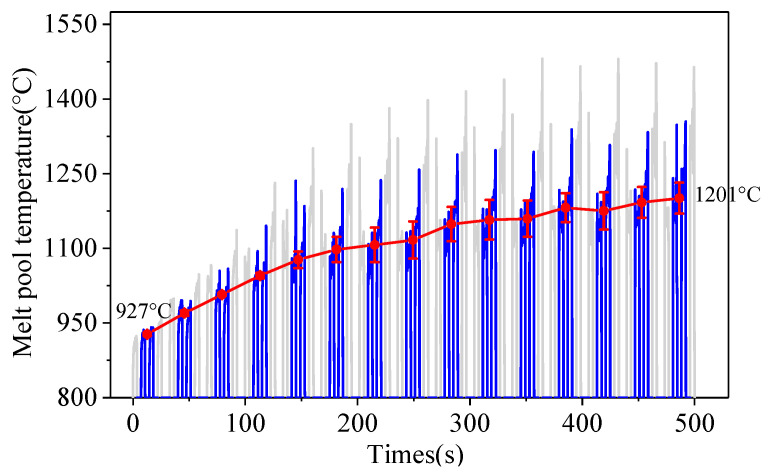
Melt pool temperature during the deposition process without a PI controller.

**Figure 7 sensors-24-05020-f007:**
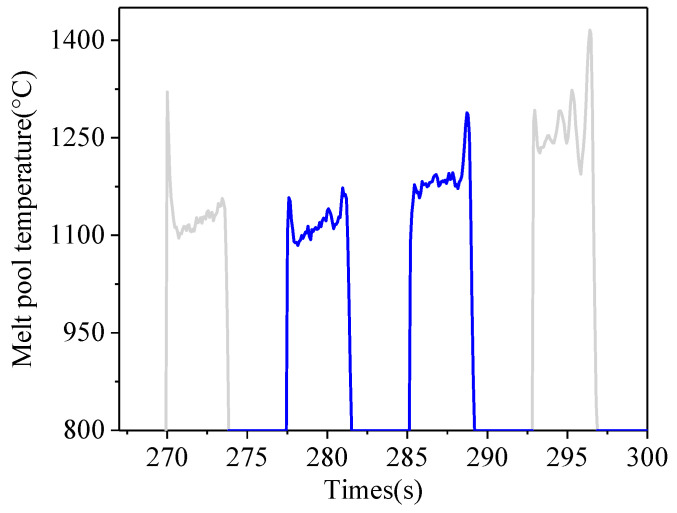
Melt pool temperature of the 9th layer.

**Figure 8 sensors-24-05020-f008:**
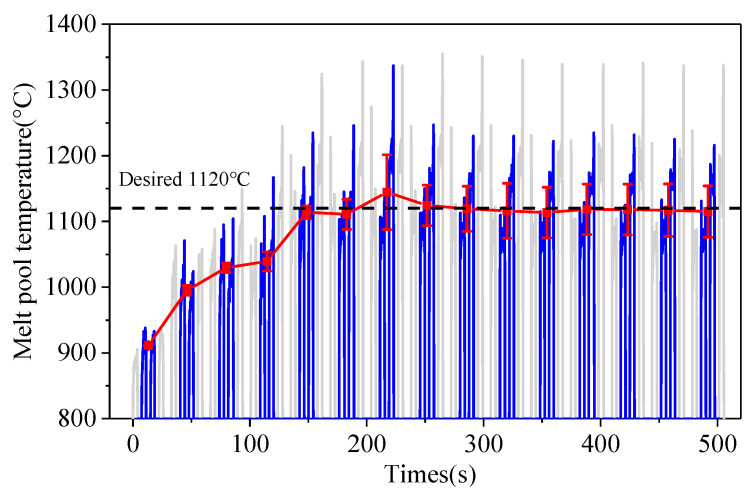
Melt pool temperature during the deposition process with a PI controller.

**Figure 9 sensors-24-05020-f009:**
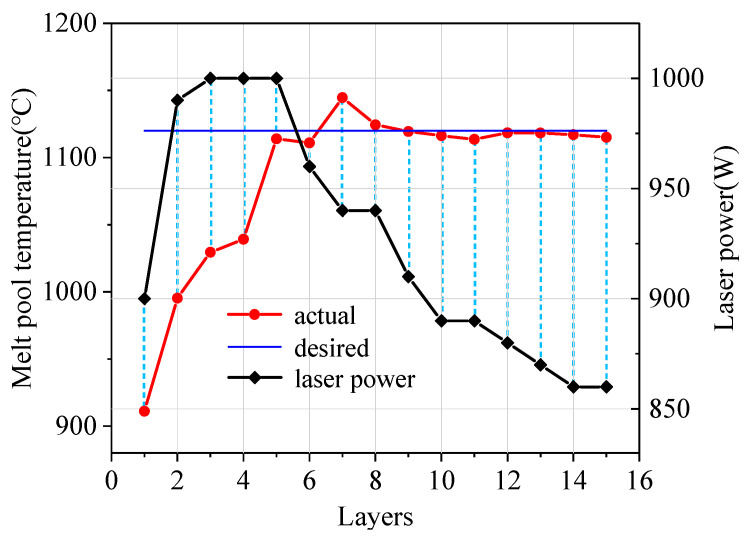
The variation of the melt pool temperature and laser power in different layers.

**Figure 10 sensors-24-05020-f010:**
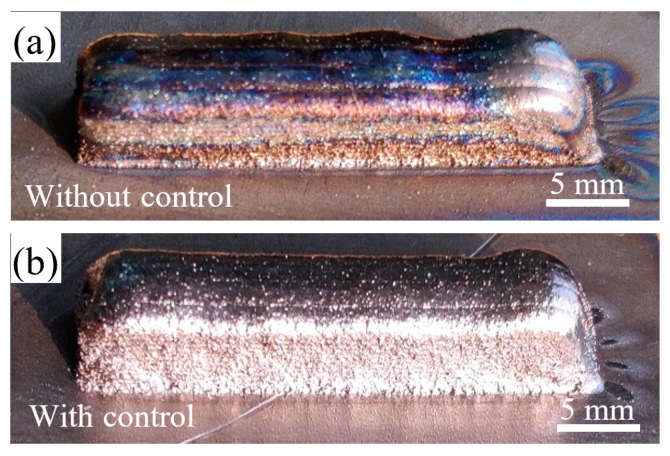
Small manufactured blocks: (**a**) without control; (**b**) with control.

**Figure 11 sensors-24-05020-f011:**
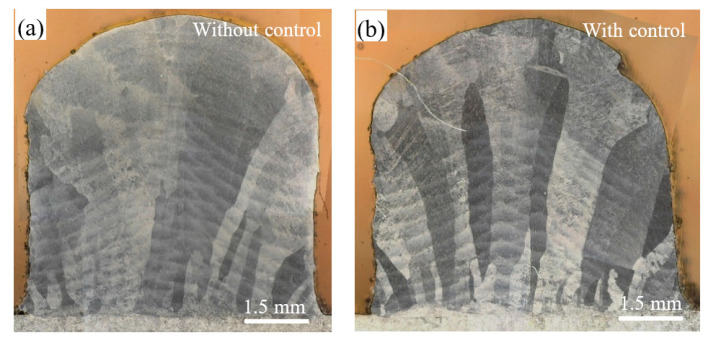
OM macrostructures of two small blocks of the cross section: (**a**) without closed-loop control; (**b**) with closed-loop control.

**Figure 12 sensors-24-05020-f012:**
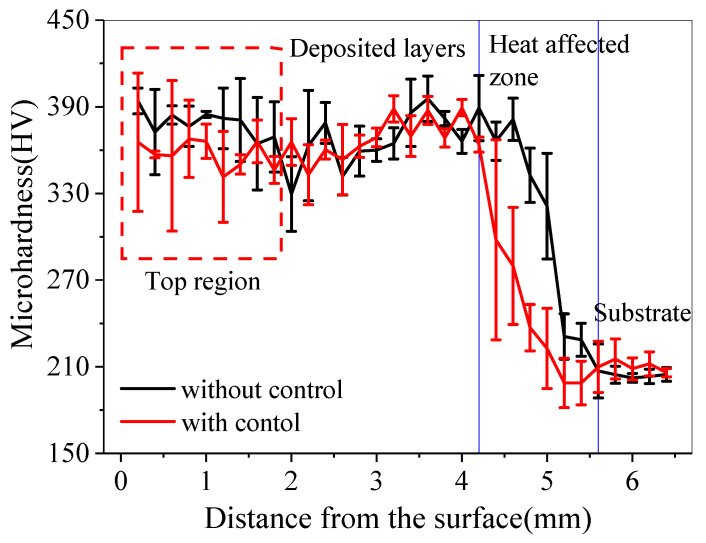
Microhardness of two small blocks with and without closed-loop control.

**Table 1 sensors-24-05020-t001:** EDS analysis of upper surface on the blocks.

	Ti	Al	V	O	N
Without control	46	5	2.6	46.3	0
With control	56.7	5.9	1.9	26.6	8.9

## Data Availability

The original contributions presented in the study are included in the article, further inquiries can be directed to the corresponding author.

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
