# Peer review of "Closed-Loop Control of Melt Pool Temperature during Laser Metal Deposition"

_sensors, 2024, doi:10.3390/s24155020_

Round 1

Reviewer 1 Report

Comments and Suggestions for Authors

no

Author Response

Dear Reviewer:

Thanks very much for your comments concerning our manuscript (sensors-3099287).

Thanks again for reviewing our paper and the valuable comments. We sincerely hope our paper will be finally acceptable and published on Sensors.

Looking forward to hearing from you soon.

Jinchao Zhang

Reviewer 2 Report

Comments and Suggestions for Authors

Process monitoring and quality control in-process for the LMD is of significance. The topic is interest. However, a lot of modifications are required to be made and the discussion should be enhanced further. A major suggestion is suggested.

1. In the introduction, the author should explain the difference from other scholars' research and elaborate on the focus of this work. It is strongly suggested that the references need to make in-depth comments on the content of the cited papers; avoid generic comments. Mention/comment the relevance of the cited paper and especially the research gap associated to it. In addition, there are more relevant papers that should be covered in literature review:

In-Situ Monitoring and Innovative Feature Fusion Neural Network for Enhanced Laser Directed Energy Deposition Track Geometry Prediction and Control

Manipulating molten pool dynamics during metal 3D printing by ultrasound

Machine vision and novel attention mechanism TCN for enhanced prediction of future deposition height in directed energy deposition 

2. What are the criteria of the monitoring angle? “The monitoring angle of the pyrometer and the CCD camera to the laser beam were 30° and 45° respectively.”

3. How do the authors control the oxidation during the deposition, especially for printing the Ti6Al4V in an open environment?

4. In Figures 9 and 10, it is suggested that label the photo so as to tell which one is deposited with control.

5. The microhardness results show that the hardness of layers with control is less than that without control, and the authors state that might be related with the Oxygen content. First, please provide the evidence to support this interpretation. Second, the Oxygen content of the condition with and without control might be same, since the deposition process was conducted in an open environment. Or is there any details to support that the increase of oxygen content in the block without closed-loop control?

Comments on the Quality of English Language

English needs to be properly improved.

Reviewer 3 Report

Comments and Suggestions for Authors

The manuscript introduces a control method of melt pool temperature for the deposition of Ti6Al4V in open environment. The melt pool temperatures with time and layers were studied. In addition, the microhardness was performed. With the controller, flatter surface and no oxidation phenomenon were obtained. However, the research contributions should be emphasized more. The mechanism for the obtained results should be discussed more. Additional research data and revisions should be added to address unclear points. Therefore, the research requires a major revision to be published in Sensors. Here are the detailed comments:

1) Some studies on temperature control in LMD were carried out. Please emphasize more the research innovations and contributions in the introduction part. In addition, some related works should be cited:

Bernauer, C., Zapata, A., & Zaeh, M. F. (2022). Toward defect-free components in laser metal deposition with coaxial wire feeding through closed-loop control of the melt pool temperature. Journal of Laser Applications, 34(4).

Petrat, T., Winterkorn, R., Graf, B., Gumenyuk, A., & Rethmeier, M. (2018). Build-up strategies for temperature control using laser metal deposition for additive manufacturing. Welding in the World, 62, 1073-1081.

Hieu, D. H., Duyen, D. Q., Tai, N. P., Thang, N. V., Vinh, N. C., & Hung, N. Q. (2022). Crystal Structure and Mechanical Properties of 3D Printing Parts Using Bound Powder Deposition Method. In Modern Mechanics and Applications: Select Proceedings of ICOMMA 2020 (pp. 54-62). Springer Singapore.

Mazzarisi, M., Angelastro, A., Latte, M., Colucci, T., Palano, F., & Campanelli, S. L. (2023). Thermal monitoring of laser metal deposition strategies using infrared thermography. Journal of Manufacturing Processes, 85, 594-611.

2) Please include the meanings of the blue and grey data in Figures 6, 7.

3) Please add the captions for Figures 9a and 9b, 10a and 10b, to clarify which one is the block with or without control.

4) The authors stated, "It is mainly related to the increase of oxygen content in the block without closed-loop control, which is consistent with the above-mentioned result that the top surface presents blue where oxidation occurs." Please include EDS measurements of the blue area and other relevant areas on the block with and without control to provide evidence for the occurrence of the oxidation process.

5) Please further discuss and explain the mechanism by which a flatter surface and lack of oxidation were achieved with the controller.

Round 2

Reviewer 2 Report

Comments and Suggestions for Authors

The paper is acceptable.

Author Response

Thanks very much for your comments concerning our manuscript (sensors-3099287). 

Reviewer 3 Report

Comments and Suggestions for Authors

The manuscript has improved a lot. However, minor revisions should be addressed to be published in Sensors. Here are the detailed comments:

1) Please correct the mistakes in lines 66 and 67. Additionally, reference 1. (after reference 22) needs to be reordered, as it is a duplicate of reference 25.:

“Devesse et al. [23Donadello et al. [A stainless steel cylinder with high precision was successfully manufactured. Devesse et al. [] introduced a control system based on a heat conduction model”

1. Donadello, S.; Motta, M.; Demir, A.G.; Previtali, B. Monitoring of laser metal deposition height by means of coaxial laser triangulation. Opt. Lasers Eng. 2019, 112, 136-144.

2) The EDS results show that the total percentage of the block without control is 99.99%, while the block with control is 91.1%. Could you please include all the chemical compositions in Table 1 and revise the explanations accordingly?
